# *Mycobacterium microti* at the Environment and Wildlife Interface

**DOI:** 10.3390/microorganisms9102084

**Published:** 2021-10-02

**Authors:** Valentina Tagliapietra, Maria Beatrice Boniotti, Anna Mangeli, Iyad Karaman, Giovanni Alborali, Mario Chiari, Mario D’Incau, Mariagrazia Zanoni, Annapaola Rizzoli, Maria Lodovica Pacciarini

**Affiliations:** 1Departemt of Biodiversity and Molecular Ecology, Research and Innovation Centre, Fondazione Edmund Mach, Via Edmund Mach 1, 38098 San Michele all’Adige, Italy; annapaola.rizzoli@fmach.it; 2National Reference Centre of Bovine Tuberculosis, Istituto Zooprofilattico Sperimentale della Lombardia e dell’Emilia Romagna, Via Bianchi 9, 25124 Brescia, Italy; mariabeatrice.boniotti@izsler.it (M.B.B.); anna.mangeli@izsler.it (A.M.); iyad.karaman@izsler.it (I.K.); giovanni.alborali@izsler.it (G.A.); mario.dincau@izsler.it (M.D.); mariagrazia.zanoni@izsler.it (M.Z.); maria.pacciarini@izsler.it (M.L.P.); 3Direzione Generale Welfare, U.O. Veterinaria, Regione Lombardia, Piazza Città di Lombardia 1, 20124 Milano, Italy; Mario_Chiari@regione.lombardia.it

**Keywords:** *Mycobacterium microti*, *Mycobacterium tuberculosis* complex, environmental samples, rodents

## Abstract

An unexpected high presence of *Mycobacterium* *microti* in wild boar in Northern Italy (Garda Lake) has been reported since 2003, but the factors contributing to the maintenance of this pathogen are still unclear. In this study, we investigated the presence of *M.* *microti* in wild rodents and in water and soil samples collected at wild boar aggregation areas, such as watering holes, with the aim of clarifying their role in *M.* *microti* transmission. In total, 8 out of 120 captured animals tested positive for the *Mycobacterium* *tuberculosis* complex (MTBC) as assessed by real-time PCR, and six samples were confirmed to be *M.* *microti*. A strain with a genetic profile similar to those previously isolated in wild boars in the same area was isolated from one sample. Of the 20 water and 19 mud samples, 3 and 1, respectively, tested positive for the presence of MTBC, and spacer oligotype SB0118 (vole type) was detected in one sample. Our study suggests that wild rodents, in particular *Apodemus* *sylvaticus*, *Microtus* sp. and *Apodemus* *flavicollis*, play roles in the maintenance of *M.* *microti* infections in wild boar through ingestion or by contact with either infected excreta or a contaminated environment, such as at animal aggregation sites.

## 1. Introduction

Mycobacterial species causing tuberculosis in humans and animals are part of the *Mycobacterium tuberculosis* complex (MTBC) [1], which includes *Mycobacterium microti*, a microorganism initially identified in England as a pathogen of wild rodents, such as field voles (*Microtus agrestis*), wood mice (*Apodemus sylvaticus*), bank voles (*Myodes glareolus*) and shrews (*Sorex araneus*) [2,3,4,5,6,7]. This pathogen causes natural infections in a wide range of wild and domestic animals, but in recent years, an increasing number of infections have been described in pets (cats and dogs) [8,9,10,11], wildlife (wild boar and badger) [12,13,14,15,16] and livestock (goat and cattle) [17,18,19,20]. In humans, *M. microti* is rarely reported as a zoonotic agent; however, its full pathogenic potential has not yet been defined [11,21]. The identification of *M. microti* by traditional methods is difficult because of its very slow growth rate in solid and liquid media [13] and the variability of its biochemical properties [22,23]. Therefore, its prevalence, geographical distribution and host range have been underestimated. Recently, owing to new approaches based on molecular methods, *M. microti* has been more readily detected and characterized. In particular, the PCR-based analysis of the IS6110 transposable element, which is present in multiple copies in *M. microti* [24], combined with the gyrB-Restriction Fragment Length Polymorphism (RFLP) assay [13], spacer oligotyping (spoligotyping) [25] and the amplification of informative chromosomal deletions, such as RD1mic and MiD1 [8,21], have been used to detect, identify and type this microorganism directly from tissue samples [7,8,12,13,15].

The presence of the MTBC in wildlife has always raised serious concerns regarding its potential epidemiological roles in infection maintenance and spread in the environment [26]. Wild boar (*Sus scrofa*) is a relevant wild species that acts as a reservoir of bovine tuberculosis (bTB) in Europe [16]. As a consequence, health-monitoring programs have been developed in several countries, including Italy [10,13,27,28,29], where wild boar habitat is expanding. Specifically, in Northern Italy (Piedmont and Lombardy regions), long-term monitoring, performed since 1998, has recorded and isolated both *Mycobacterium bovis* and *M. microti* in wild boar [13,15,28,30,31]. The most extensive study, over a 9-year period (2003 to 2011), was performed in Lombardy (Central Alps, Italy) by Boniotti et al. (2014) [13] in which approximately 23,000 hunter-harvested head lymph nodes were examined using a molecular approach. An unexpectedly high presence of *M. microti* was found, whereas *M. bovis* was identified in only two animals. This widespread presence of *M. microti* infections in wild boar has also been described by Chiari et al. (2016) [15], and they confirmed the presence of a “hotspot” zone (prevalence of 6.2%) in a hunting area characterized by a footstep mountain habitat located to the north of Garda Lake (Gargnano Municipality, Brescia Province, Italy). These data support the hypothesis that wild boar play an active role in maintaining *M. microti* in the environment, but many related factors still need to be investigated, such as the epidemiological cycle at the intra- and interspecific levels and the transmission of this microorganism in the environment. The circulation of MTBC bacteria within multi-host systems, including cattle and various wild species, such as badgers (*Meles meles*), wild boar (*S. scrofa*) or red deer (*Cervus elaphus*), has been well documented [32,33,34,35]. Infected animals can excrete *M. bovis* via sputum, feces, urine and any type of aerosol [36,37,38], thereby contaminating the environment [36], where the bacteria have long survival times [39,40]. Thus, the data strongly suggest that other components of the MTBC, such as *M. microti*, persist in the environment and are transmitted to other susceptible host species, such as wild boar and small rodents. Smith et al. (2009) [8] suggested that domestic cats are spillover hosts (sentinels) owing to the presence of the bacteria in local voles, which maintain *M. microti* in certain endemic areas of Great Britain. It is reasonable to hypothesize that cats contract *M. microti* when hunting infected small mammals and that the pathogen can also be present in the environment and in other wild species, such as badgers, that are very common in these areas. Demonstrating the presence of *M. microti* using culture methods is difficult because of its slow growth rate and, in abiotic samples, owing to the abundance of soil microbial communities. To circumvent these drawbacks, several new procedures based on a combination of molecular approaches, with pre-treatment steps for concentrating mycobacteria from water and soil substrates and removing PCR inhibitors, have been described [35,41,42,43].

Thus, the objectives of this study were to assess the presence of *M. microti* in wild rodents and at wild boar aggregation sites in the “hot spot” area identified previously to the north of Garda Lake [13,15]. To the best of our knowledge, this is the first study to undertake monitoring for the presence of *M. microti* in a wild rodent community and in environmental samples from an area where its presence in wild boar has been well documented.

## 2. Materials and Methods

### 2.1. Study Area and Sample Collection

The study area is an Officially Tuberculosis-free region (OTF) located in Brescia Province, Italy, included in the ‘Parco dell’Alto Garda Bresciano’, north-west of Lake Garda. The territory spans from the Mediterranean climate along the shores of Lake Garda, at 65 m a.s.l., to the alpine climate of the highest peak, Mount Caplone, at nearly 2000 m a.s.l. The provincial territory is divided into nine hunting districts in which culled wild boars have been monitored by the Istituto Zooprofilattico della Lombardia e dell’Emilia Romagna since 2003. *Mycobacterium microti* has been continuously detected in this species and has increased markedly in the population [13,15].

Wild rodent and environmental samples were collected from a specific hunting zone where the highest prevalence of the pathogen in the wild boar population has been recorded [15]. Small mammals were trapped in 2016 and 2017 using snap traps placed along transects located in habitats where wild boar spend significant time, such as ecotones and meadows, or where they aggregate, such as ponds. In 2016, the sampling was carried out at five trapping sites in June and September, whereas in 2017, nine sites were monitored in July, August and September (Appendix A, Figure 1). At each session, 250 traps were activated and checked daily for 3–4 days. Animals were individually placed in tubes and transported to the laboratory in a portable freezer to avoid tissue deterioration. In the laboratory, animals underwent necropsies and dissection. The liver, spleen, heart and lung were pooled and stored at −80 °C until analyzed. All trapping and sampling procedures were approved by the Wildlife Committee of the Autonomous Province of Trento (Prot. n. S044-5/2015/277268/2.4).

In 2017, environmental samples were collected from nine watering holes. In particular 1–1.5 l of water and 1–1.5 kg of mud were collected at the beginning and end of the sampling season, resulting in 36 samples (18 of water and 18 of mud. Samples 12 and 17 (water) and sample 12 (mud) were collected in duplicate). Each sample was individually stored at −80 °C by site and date in sealed plastic bags until processed and analyzed.

### 2.2. Culture Isolation and DNA Extraction of Tissue Samples

Rodent samples were subjected to culture isolation and DNA extraction in accordance with the procedures described previously [13], with minor modifications. The preparation of homogenates was performed using approximately 3–5 g of tissue sample in 6–10 mL 1× phosphate-buffered saline (PBS, pH 6.8) (30 mM phosphate buffer at pH 7.2, 150 mM NaCl and 2 mM EDTA), with a 1:2 weight:volume ratio, in a disposable mechanical disgregator Medicons mixer (Medimachine-Medimax) for 180 s. Sample aliquots used for DNA extraction were inactivated at 98 °C for 10 min. DNA extraction was performed using a Nucleospin tissue kit (Macherey Nagel) in accordance with the manufacturer’s recommendations. DNA was eluted in a final volume of 300 μL of TE buffer (10 mM Tris-HCL and 1 mM EDTA at pH 8.0).

### 2.3. Experimental Inoculation and DNA Extraction of Abiotic Samples

A suspension of the *M. bovis* BCG (ATCC 27290) inoculum was created as follows: a 21-d-old *M. bovis* BCG liquid culture was vortexed for 60 s to disrupt the aggregates. Approximate 10-fold dilutions (10^−1^ to 10^−6^) were prepared in saline solution. The *M. bovis* cell concentrations were estimated by spreading duplicate 100 μL aliquots of 10^−3^, 10^−4^ and 10^−5^ diluted suspensions onto Middlebrook 7H11 agar enriched with 10% Oleic Albumin Destrose Catalase growth supplement (OADC). Plates were incubated at 37 °C for 30 days, and then, *M. bovis* colonies were counted. The titer of the *M. bovis* BCG suspension was calculated as 3.6 × 10^8^ Colony-Forming Unit (CFU) per ml. 

Uninoculated mud or water used to prepare experimental contaminations tested negative in preliminary experiments with IS6110 targeted real-time PCR (IS6110 RT-PCR) and with a VetMAX *M.* tuberculosis complex kit.

### 2.4. Water Samples

To assess each protocol, 9.0 mL aliquots of sterile water were spiked in duplicate with 1 mL of 10-fold serial dilutions of the inoculum suspension to reach *M. bovis* BCG concentrations ranging from 3.6 × 10^6^ CFU·ml^−1^ to 3.6 × 10 CFU·ml^−1^.

Replicate set samples were passed through a 0.45 μm nitrate Millipore filter using a vacuum pump (Speed-Flow). Filters were broken down for 180 s in a Stomacher 80 laboratory blender. Afterwards, samples were resuspended in 8 mL of PBS. They were then inactivated and mechanically lysed using 100 μg of glass beads (100 to 200 μm in diameter) as described previously [13]. The final concentrations of *M. bovis* BCG·ml^−1^ after the first step of sample processing ranged from 4.5 × 10^5^ to 4.5 × 10 CFU·ml^−1^.

DNA was extracted from 300 μL aliquots of contaminated water samples using the following protocols:
Extraction by affinity spin columns using a Nucleospin tissue kit (Macherey-Nagel, Düren, Germany) in accordance with the manufacturer’s procedures; andSemi-automatic extraction by magnetic beads using a MagMax Core Nucleic Acid purification kit (Applied Biosystems, ThermoFisher Scientific, Inc., USA) with an automatic extractor (KingFisher Flex, Applied Biosystem, ThermoFisher Scientific, Inc., USA) following the manufacturer’s recommendations.


In both cases, DNA samples were eluted in 100 μL final volumes of TE buffer (10 mM Tris-HCL and 1 mM EDTA at pH 8.0).

### 2.5. Mud Samples

To optimize sample preparation and DNA extraction, 19 g aliquots of urban mud samples were randomly chosen in the Italian bTB-free areas (Brescia, Italy). Samples were spiked in duplicate with 1 mL of 10-fold serial dilutions of *M. bovis* BCG (ATCC 27290) inoculum suspensions (3.6 × 10^7^ to × 10^1^ CFU·g^−1^) obtaining final concentrations of 1.8 × 10^6^ to 1.0 × 10 CFU·g^−1^ The samples were inactivated and mechanically lysed as described previously [13] and then DNA was extracted using four different procedures. In procedures 2 and 4, mechanical lysis was performed by vortexing horizontally at a high speed for 2.5 min using a Vortex Genie 2 (Scientific Industries Inc., New York, NY, USA). The details of the four procedures are as follows:

Procedure 1: as suggested by the manufacturer of the MagMAX Core Mechanical lysis manual for the DNA extraction of *Mycobacterium avium* subspecies *paratuberculosis* in feces (Applied Biosystems, Thermofisher Scientific, Inc., USA), 3 g of initial suspensions were diluted in 20 mL of sterile water. Then, 1.8 mL aliquots, corresponding to the presence of 4.8 × 10^5^ CFU, and dilutions up to 4.8 × 10^1^, were subjected to semi-automatic extraction with magnetic beads using a MagMax Core Nucleic Acid purification kit (Applied Biosystems, ThermoFisher Scientific, Inc., USA) and the automatic extractor (KingFisher Flex, Applied Biosystem, ThermoFisher Scientific, Inc., USA) following the manufacturer’s recommendations. DNA samples were eluted in 200 μL of TE buffer.

Procedure 2: In total, 5 g of initial suspensions, corresponding to the presence of 9 × 10^6^ CFU, and dilutions up to 9 × 10^1^ CFU of *M. bovis* BCG, were extracted using affinity spin columns and the DNeasy PowerMax Soil kit (Qiagen, Germany) in accordance with the manufacturer’s procedures. DNA samples were eluted in 5000 μL of elution buffer.

Procedure 3: In accordance with the method described by Yamanouchi et al. (2018) [43], 3 g of initial suspensions, corresponding to the presence of 5.4 × 10^6^ CFU, and dilutions up to 5.4 × 10^1^ CFU of *M. bovis* BCG, were extracted with phenol-chloroform. Powdered milk was added to eliminate contaminants. DNA samples were eluted in 600 μL of TE buffer.

Procedure 4: In accordance with the method described by Thorn et al. (2018) [42], 2 g of initial suspensions, corresponding to the presence of 3.6 × 10^6^ CFU and dilutions up to 3.6 × 10^1^ CFU of *M. bovis* BCG, were extracted with hexadecyltrimethyl-ammonium bromide (CTAB) buffer (2% CTAB, 1.4 M NaCl, 100 nm Tris-HCl and 20 mM EDTA, pH 8) after a pretreatment with a three polar positive amino acid mix (arginine, histidine and lysine). Samples were purified with phenol-chloroform and precipitated with PEG 6.000. DNA samples were eluted in 100 μL of TE buffer.

### 2.6. DNA Extraction and Purification of Environmental Samples from the Field

For water samples, the microbial biomass was concentrated by filtering all the sample (1–1.5 L) in 3–4 steps, depending on the turbidity level. Water was poured initially through a coffee filter and subsequently forced using a vacuum pump through cellulose MilliporeTM filters having various pore sizes (2.5 μm, 1 μm and 0.45 μm). Millipore filters were inactivated and broken down as previously described in the section “Experimental inoculation and DNA extraction of abiotic samples”. DNA extraction was performed using Protocol B (see Results).

For soil, 2–5 g aliquots were randomly collected from each sample for subsequent analyses. DNA extraction was performed following Protocol 1 (see Results).

### 2.7. PCR Reactions and Sequence Analysis

All the PCR reactions were performed using 5 μL DNA aliquots extracted from water, soil or tissue samples. For the detection of the MTBC, IS6110 RT-PCR was performed in accordance with the procedure described previously [13], and 0.5 μL of the Internal Control (IC, Quantifast pathogen kit, Qiagen, Germany) was added to each sample before DNA extraction. Positive tissue samples were confirmed by amplification with the VetMAX *M.* tuberculosis complex kit (Applied Biosystems, ThermoFisher Scientific, Inc., USA) in accordance with the manufacturer’s instructions.

The molecular identification and characterization of the presence of the MTBC and *M. microti* were performed directly on DNA samples and on the *M. microti* isolates using a combination of the following reactions: PCR reactions described by Kulski et al. (1995) [44] for the identification of *Mycobacterium* spp. and *M. avium* (Kulski reactions); PCR-RFLP assays of the gyrB gene for the identification of *M. microti* as described by Boniotti et al. (2014) [13]; detection of the RD1mic region for the identification of *M. microti* as described by Smith et al. (2009) [8]; spoligotyping, multilocus variable number tandem repeat (MLVA) typing with 12 VNTRs (ETRA, ETRB, ETRC, ETRD, ETRE, Qub11a, Qub11b, Qub26, Qub1895, Qub15, VNTR3232 and MIRU26) as described by Boniotti et al. (2009) [45]; and sequencing a portion of the Direct Repeat locus as described previously [13].

*Mycobacterium* other than *tuberculosis* were characterized by sequencing the 1030 bp amplicons obtained by the Kulski PCR reaction for the identification of *Mycobacterium* spp. using its specific primers and the Big Dye Terminator vers.1.1. Cycle sequencing Kit (Applied Biosystems, ThermoFisher Scientific, Inc., USA) following the manufacturer’s recommendations. Sequencing was performed in an ABI Prism 3500 XL genetic analyzer (Applied Biosystems, ThermoFisher Scientific, Inc., USA).

## 3. Results

### 3.1. Screening of Wild Rodents 

Between 2016 and 2017, 120 animals were trapped and culled; 75 (62.5%) were yellow-necked mice (*Apodemus flavicollis*), 30 (25%) were wood mice (*A. sylvaticus*), 13 (10.8%) were voles (*Microtus* sp.) and 2 were (1.7%) hazel dormice (*Moscardinus avellanarius*) (Appendix A). None of the examined carcasses revealed the presence of macroscopic lesions.

Tissue pools yielded eight positive results (six *A. flavicollis*, one *A. sylvaticus* and one *Microtus* sp.) for the MTBC using IS6110 RT-PCR and the *M. tuberculosis* complex VetMAX kit. Six and five of the eight positive samples, were confirmed to be *M. microti* using the gyrB RFLP assay and RD1mic region detection, respectively (Table 1), whereas only one was isolated using culturing protocols with a prolonged incubation time (18 weeks). Details of the results for each animal are reported in the Appendix A.

The eight positive animals were trapped in the localities of Sembrune (3) and Bertù (5) (Figure 1), resulting in two and four identified *M. microti* cases, respectively, in 2017. The isolation of *M. microti* was possible only from *Microtus* sp. No positivity for the MTBC was detected in the other sites. 

Genotyping using spoligotyping and MLVA was performed on the eight MTBC positive samples (Table 1). No conclusive results were obtained when spoligotyping was directly applied to DNA from the tissue samples. However, it was possible to identify the presence of SB2277, which is characterized by the absence of spacers, and a large MiDi deletion [13] was observed by sequencing a portion of the RD region in an *M. microti* isolate (sample 67). The RD1mic region was also used for characterization and was detected in DNA from tissues of samples 68, 69, 71 and 72.

The VNTR analysis identified genotype 9,3,5,6,1,5,9,9,4,3,14,2 (order of markers as described in the Materials and Methods), whereas partial results were obtained directly in the DNA from tissues of samples 68, 69 and 71 (see Table 1).

Genotype SB2277 combined with ETRA-E typing (9, 3, 5, 6, 1) has been previously described [13] in *M. microti* strains isolated from wild boar in the same locality.

### 3.2. Comparison of Procedures for Environmental Sample Analysis

To select the most accurate and feasible method for detecting the MTBC, *M. microti* and other mycobacteria present in the environmental matrices, we evaluated different lysis and DNA extraction procedures using experimentally contaminated water and soil samples. 

For water, the results obtained using the two different protocols (see Materials and Methods) are presented in Table 2. Procedure B (Semi-automatic extraction by magnetic beads) proved to be more efficient than Protocol A and was able to detect up to the fourth 10-fold dilution (limit of detection in 300 μL processed substrate, 1.5 × 10^2^), corresponding to the presence of approximately seven genomic copies in the 5 μL DNA sample analyzed by RT-PCR.

For soil, we compared four protocols having different combinations of lysis and DNA extraction procedures (see Materials and Methods). The results, presented in Table 3, show that Protocols 1 and 2 (Semi-automatic extraction by magnetic beads and Qiagen kit, respectively) detected up to the second 10-fold dilution (4.5 × 10^3^ and 9 × 10^4^, respectively), corresponding to the presence of approximately 1200 and 900 genomic copies, respectively, in the final volume analyzed by RT-PCR. Although their efficiency levels were similar, Protocol 1 had a better feasibility and a greater processivity than Protocol 2. Protocols 3 and 4 described by Yamanouchi et al. (2018) [43] and by Thorn et al. (2018) [42], respectively, produced negative results for all the dilutions, indicating that these procedures failed in the recovery of DNA from mycobacteria and/or in the elimination of inhibitors.

### 3.3. Molecular Analysis of Environmental Field Samples

For the analysis of water samples collected from watering holes (Figure 1), we used Protocol B (see section “Comparison of procedures for environmental samples analysis”). DNA samples were subjected to IS6110 RT-PCR. Three MTBC-positive samples were detected (Table 4), two at Bocca di Lovere (3 and 4) and one at Praa 1 (13) sites. Further molecular tests (gyrB PCR-RFLP assays) were performed on the positive samples without obtaining any amplification of the target, with the exception of the spoligotyping of sample 3, which showed the presence of SB0118, the characteristic vole-type profile of *M. microti*. All the samples underwent Kulski PCR reactions. In total, 14 samples tested positive by PCR for *Mycobacterium* spp. The sequencing of these PCR products revealed the presence of *Mycobacterium* chelonae in 12 of the 14 positive samples. In particular, in sample 3, both *M. microti* (positive for the MTBC and spoligotyping) and *M.*
*chelonae* were detected.

For the analysis of the 19 mud samples collected at watering holes (Figure 1), we selected Protocol 1 (Semi-automatic extraction by magnetic beads) on the basis of the trial results (see section “Comparison of procedures for environmental samples analysis”). Of the samples subjected to IS6110 RT-PCR, one tested positive (4). It had been collected at the Bocca di Lovere site. Further molecular tests (PCR-RFLP gyrB and spoligotyping, Table 5) did not produce any positive or interpretable results. The amplification of the IC showed the partial inhibition of samples 1, 9, 10, 11, 13, 14 and 16.

## 4. Discussion

Previous studies revealed an unexpected high presence of *M. microti*, along with the presence of macroscopic lesions compatible with tuberculosis, in wild boar in Italy, particularly in Gargnano, in the Alto Garda Bresciano region [13,15]. These data support the hypothesis that wild boar plays an active role in the transmission of this microorganism; however, factors that contribute to the maintenance of *M. microti* in the environment have not been thoroughly investigated. Here, we aimed to assess the presence of small mammals in the Gargnano area and perform a preliminary screening for the presence of *M. microti* in both rodents and environmental substrates, such as water and mud, collected at wild boar aggregation sites. 

Several studies, in particular from the UK, have shown that the main natural reservoirs of *M. microti* are field voles, bank voles and shrews [5,6,7,8,21]. Other indirect studies provide evidence of the presence of *M. microti* in field voles through infections of cats [8,9]; however, in the work of Peterhans et al. (2020) [11], which describes 11 cases of *M. microti* in cats in Switzerland, 346 wild mice (*Microtus* agrestis) captured in the presumptive endemic area failed to demonstrate the presence of this microorganism. Recently, Perez de Val et al. (2019) [19] and Ghielmetti et al. (2020) [16], reported outbreaks of *M. microti* in wild boar in Spain and Switzerland, respectively, but no specific investigations were conducted on the possible roles of other susceptible species or of the environment in the *M. microti* epizootiological cycle.

In our study, the area included a fragmented habitat with forest patches, meadows and pastures. Due to the ecological requirements of the natural rodent reservoir cited in the literature, our main target habitats were open areas, such as meadows and ecotones. Nonetheless, the dominant species captured was the yellow-necked mouse (*A. flavicollis*) which, although preferring forested habitats from sea level to above the tree line, can occupy many niche environments. We also captured species typical of open areas, such as wood mice (*A. sylvaticus*) and voles. The number of animals collected in the first year was lower (35% of the total) than in the second, and no positive animals were detected. This could be because of the scarcity of samples, but also the choice of monitoring sites. In fact, in the second year, trapping sites were selected in the proximity of wild boar watering areas, and these produced some positive results. In total, 8 out of 120 captured animals were positive for the MTBC as assessed by IS6110 RT-PCR. The PCR/RFLP gyrB assay and the RD1mic PCR confirmed the presence of *M. microti* in six and five animals, respectively. However, the territory being OTF and the high prevalence of *M. microti* described previously in this area [13,15] lead us to believe that all eight MTBC-positive animals were infected by this pathogen. *Mycobacterium microti* was isolated from sample 67 (*Microtus* sp.) and genotyped. Its genetic profile, SB2277 combined with ETRA-E typing (9,3,5,6,1) was described previously in strains isolated in wild boar in 2007, 2009 [13], 2012, 2013 and 2017 in the hunting area of Gargnano (Boniotti, personal communication). The presence of SB2277, characterized by the absence of spacers and by a large MiD1 deletion, is reported in voles for the first time in this work through the sequencing of the Direct Repeat locus of a strain isolated from *Microtus* sp. The other seven animals positive for the MTBC as assessed by RT-PCR (including the two assumed to be *M. microti*) were *A. flavicollis* and *A. sylvaticus* in case six and one, respectively. To the best of our knowledge, this is the first reported detection of *M. microti* in *A. flavicollis*. Although the isolation of *M. microti* was not possible, its presence was confirmed in samples 68, 69, 71, 72 and 73 using molecular tests. The partial genotypes obtained with MLVA of samples 68, 69 and 71 matched the genotype found in sample 67 and those described previously in wild boar from the same area [13].

The importance of the environment in the maintenance of the natural cycle of *M. microti* is supported by the following: (i) mycobacteria are common inhabitants of the soil. The majority are innocuous, but some infect humans and other animals [46]; (ii) infected soils shared between sympatric wildlife and livestock may become key zones for the indirect transmission of mycobacteria. In this way, they may colonize small mammal tissues or simply pass through their digestive systems and be shed intact in feces and bodily fluids; and (iii) the main recognized risk factor for bTB maintenance and spread worldwide appears to be the gathering of individuals triggered by a central point of attraction in any localized area. In our case, this was represented by watering areas. 

We analyzed samples (water and mud) collected from watering holes used by wild boar and wildlife. Our results confirmed, as in the literature, that the application of molecular methods to environmental samples is difficult, in particular for soil/mud. Due to its physical–chemical characteristics, soil/mud may contain several inhibitors, such humic and fulvic acids [41], and other specific environmental contaminants, such as heavy metals and phenolic compounds [47]. Additionally, and intrinsically linked with the first problem, is the difficulty of using a representative portion of the sample [35,41,42,43]. To optimize the DNA extraction procedures for *M. bovis* BCG experimentally contaminated water and soil samples, first we evaluated different protocols selected from the scientific literature and available commercial kits. For mud, because it is a problematic substrate, we decided to assess four protocols based on different principles of DNA extraction and inhibitor elimination. However, Procedures 1 and 2 failed to recover the target DNA and/or remove the inhibitors. These are laborious, time-consuming methods that require several steps, which may have contributed to the loss of DNA. The other procedures (3 and 4) detected up to the 4.8 × 10^3^ CFU dilution, corresponding to the presence of hundreds of genomic copies in the amplified sample. However, Procedure 3, based on the use of magnetic beads and semi-automatic extraction, had a better feasibility and a greater processivity. Consequently, it was chosen for subsequent analyses.

The application of the selected protocols to field environmental samples resulted in the detection of the MTBC in three water samples collected from two sites and one mud sample. Two of the positive water samples (3 and 4) were collected from the same location (Bocca di Lovere) at different times (July and September, respectively), highlighting the persistence of this microorganism. The only positive mud sample (4) was retrieved from the same site (Bocca di Lovere; site 10 in Figure 1) at the end of the trapping season. *M. microti* could not be identified in any of the environmental samples using molecular methods, with exception of the one water sample (3), where it was detected as the presence of SB0118, a spoligotype commonly described as “Vole Type” [24]. Interestingly, the positive rodent samples were located 0.5 (Sembrune site) and 1.5 (Bertu site) km away, confirming the presence of the MTBC in this wild boar aggregation area and the possible involvement of environmental factors. The other positive water sample was located 2 km away (Praa1 site).

Moreover, the application of Kulski PCR for *Mycobacterium* spp. identification and sequencing allowed us to identify the presence of *M.* chelonae in 9 out of 20 samples. *Mycobacterium* chelonae is a ubiquitous *Mycobacterium* that is widespread in fresh water sources and aquatic animals. It has been described as pathogen of fish, but it has also been isolated in association with granulomatous diseases of snakes, turtles [48] and mice [49]. It has also been occasionally found in cattle [50], pigs [51,52,53], cats [54] and dogs [55].

All the collected data confirmed the hypothesis that some genotypes of *M. microti* are circulating among voles and wild boar, and probably also through transmission in the environment. It was possible to confirm the presence of *M. microti* in only one water sample by spoligotyping (SB0118). However, because the study was performed in an OTF footstep mountain habitat where cattle herds are not present and *M. microti* has been continuously detected in wild boar since 2007, we can reasonably assume that the positivity to the MTBC corresponds in all cases to the presence of *M. microti*.

The wild boar is a highly versatile omnivore species. Together with nuts, berries and seeds, the majority of its diet consists of food items dug from the ground, including burrowing animals, such as earthworms, insects, rodents and insectivores. Wild small mammals could spread the disease to their predators, as in the UK, with cats eating infected voles [9], or by contaminating the water or soil with their excreta. Infected wild boar themselves may contribute to the spread of *M. microti* during their aggregation at watering areas. In the work of Smith et al. (2009) [8] and Peterhans et al. (2020) [11], it was speculated that the natural transmission of the bacteria from wild rodents (considered the maintenance host of *M. microti*) to cats might have occurred through the oral infection route and that felines may be considered a spillover host. In our context, it is important to remember that most of the positive wild boar reported in previous studies [13,15] showed tuberculosis-like lesions in retropharyngeal and mandibular lymph nodes and sometimes in mesenteric lymph nodes (data not shown), suggesting that they are not just dead-end hosts but can play roles in the active transmission of this microorganism in the environment and in other host species. An additional consideration is that the detection of *M. microti* in wild boar was performed over a long time (more than 10 years) on 1000 animals during the bTB monitoring program, whereas *M. microti* was detected in voles after analyzing a small population (120 animals) in 2 years from precise monitoring sites. This may indicate that the presence of this microorganism in these host species is much more widespread than detected in this study.

We cannot exclude, on the basis of current knowledge and our results, that all these factors (intra- and interspecies transmission and environment spread) play active roles in maintaining the circulation of *M. microti* in this endemic area. Further studies are required to better characterize the epidemiology of this mycobacteria and the possible implications for the ecosystem and animal health.

## 5. Conclusions

*Mycobacterium microti* has been recorded in an OTF area in Northern Italy, both in wild boar from 2011 to 2017 and in rodents and environmental samples in 2017. The similarity of the genotypes and the positivity to the MTBC confirms the interspecies circulation of this *Mycobacterium* and its presence in the environment. We believe that these factors are involved in the maintenance and transmission of *M. microti* in this endemic area, although further studies are necessary to assess with certainty the precise roles of all the players in the infection cycle of this pathogen. In this study, the presence of genotype SB2277 in a *M. microti* strain isolated from *Microtus* sp. was demonstrated, moreover *A. flavicollis* was identified as a new susceptible species to this pathogen.

## Figures and Tables

**Figure 1 microorganisms-09-02084-f001:**
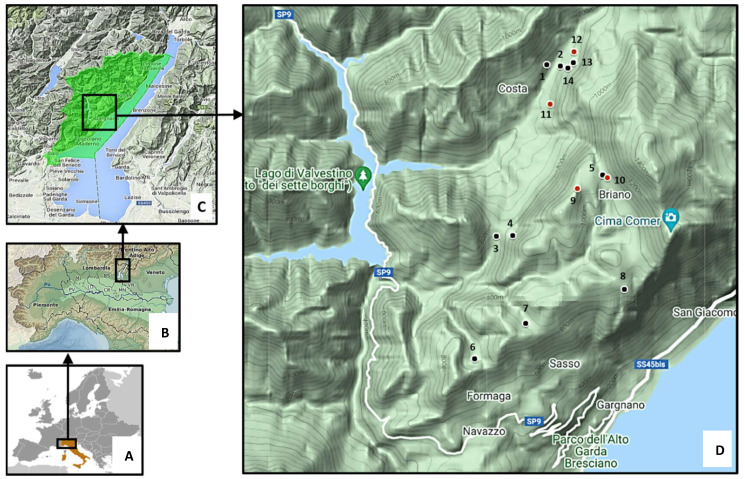
Map of the study area: (**A**) = Italy with the macro region area denoted; (**B**) = Lombardy region and Lake Garda area, BS represents Brescia Province; (**C**) = Alto Garda Bresciano Park area denoted in green; (**D**) = Gargnano area with study sites. Red circles indicate sites positive for *M. microti* (9 and 11 positive rodent samples; 10 and 12 positive environmental samples). 1 = Via Costa 1; 2 = Via Costa 2; 3 = Passo di Magno 1; 4 = Passo di Magno 2; 5 = Briano; 6 = Lama; 7 = Navone; 8 = Fa; 9 = Sembrune; 10 = Bocca Lovere; 11 = Bertu; 12 = Praa 1; 13 = Praa 2; 14 = Praa 3. Further details on the sites can be found in Appendix A.

**Table 1 microorganisms-09-02084-t001:** Molecular typing results of MTBC-positive rodent tissue samples (P = positive; N = negative, NC = not conclusive, NA = not amplifiable).

ID	Species	Site	Culture Results	RT-PCR IS6110	PCR/RFLP *gyrB* Assay	RD1^mic^	Spoligotyping	MLVA Typing
67	*Microtus* sp.	Sembrune	P	P	*M. microti*	P	SB2277	9,3,5,6,1,5,9,9,4,3,14,2
68	*A. flavicollis*	Sembrune	N	P	*M. microti*	P	NC	NA,3,5,6,1,5,9,9,4,3,14,2
69	*A. flavicollis*	Bertù	N	P	*M. microti*	P	NC	NA,3,5,6,1,5,9,9,4,3,14,2
71	*A. flavicollis*	Bertù	N	P	*M. microti*	P	NC	NA,3,5,NA, 1,5,9,9, 7,3,14,2
72	*A. flavicollis*	Bertù	N	P	*M. microti*	P	NC	NA
73	*A. flavicollis*	Bertù	N	P	*M. microti*	N	NC	NA
74	*A. sylvaticus*	Bertù	N	P	NA	N	NC	NA
78	*A. flavicollis*	Sembrune	N	P	NA	N	NC	NA

**Table 2 microorganisms-09-02084-t002:** IS6110 RT-PCR results, expressed in cycle threshold (Ct), of experimentally contaminated water samples (C− = negative control; P = positive; N = negative; NA = not amplified). * Ct values were calculated as the average of Ct replicate values.

*M. bovis* CFU	Procedure A Ct * Value	Result	Procedure B Ct * Value	Results
4.5 × 10^5^	26.05	P	24.27	P
4.5 × 10^4^	31.25	P	28.78	P
4.5 × 10^3^	38.00	P	30.50	P
4.5 × 10^2^	NA	N	37.2	P
4.5 × 10^1^	NA	N	NA	N
C−	NA	N	NA	N
C−	NA	N	NA	N

**Table 3 microorganisms-09-02084-t003:** IS61160 RT-PCR results, expressed in cycle threshold (Ct), of experimentally contaminated mud samples (C− = negative control; P = positive; N = negative; NA = not amplified). * Ct values were calculated as the average of Ct replicate values.

*M. bovis* BCG CFU	Protocol 1 Ct * Value	Result	*M. bovis* BCG CFU	Protocol 2 Ct * Value	Results
4.8 × 10^4^	P (30.5)	P	9 × 10^5^	P (29)	P
4.8 × 10^3^	P (35.5)	P	9 × 10^4^	P (33)	P
4.8 × 10^2^	NA	N	9 × 10^3^	NA	N
4.8 × 10^1^	NA	N	9 × 10^2^	NA	N
C−	NA	N	C−	NA	N
C−	NA	N	C−	NA	N

**Table 4 microorganisms-09-02084-t004:** Analysis of water samples from the study sites (P = positive, N = negative, NP = not performed, NC = not conclusive).

Sample ID	Site	Date of Collection	PCR ^1^	PCR ^2^	PCR ^3^	Sequence Identification	PCR-RFLP *gyrB* Assay	Spoligotyping
1	Bertù	25 July 2017	N	N	N	NP	NP	NP
2	Bertù	20 June 2017	N	N	P	*M. chelonae*	NP	NP
3	Bocca di Lovere	25 July 2017	P	N	P	*M. chelonae*	N	SB118
4	Bocca di Lovere	20 September 2017	P	N	N	NP	N	NC
5	Fa	25 July 2017	N	N	P	*M. chelonae*	NP	NP
6	Fa	20 September 2017	N	N	P	*M. chelonae*	NP	NP
7	Lama	26 July 2017	N	N	P	NC	NP	NP
8	Lama	21 September 2017	N	N	P	NC	NP	NP
9	Navone	25 July 2017	N	N	P	NC	NP	NP
10	Navone	20 September 2017	N	N	P	*M. chelonae*	NP	NP
11	Praa 1	25 July 2017	N	N	P	*M. chelonae*	NP	NP
12	Praa 1 BIS	26 July 2017	N	N	P	*M. chelonae*	NP	NP
13	Praa 1	21 September 2017	P	N	N	NP	N	NC
14	Praa 2	26 July 2017	N	N	P	*M. chelonae*	NP	NP
15	Praa 2	21 September 2017	N	N	P	*M. chelonae*	NP	NP
16	Praa 3	25 July 2017	N	N	P	*M. chelonae*	NP	NP
17	Praa 3 BIS	26 July 2017	N	N	P	*M. chelonae*	NP	NP
18	Praa 3	21 September 2017	N	N	P	*M. chelonae*	NP	NP
19	Sembrune	25 July 2017	N	N	N	N	NP	NP
20	Sembrune	20 September 2017	N	N	N	N	NP	NP

^1^ = IS6110 RT-PCR. ^2^ = PCR Kulski *M. avium.* ^3^ = PCR Kulski *Mycobacterium* spp.

**Table 5 microorganisms-09-02084-t005:** Analysis of mud samples from field sites (P = positive, N = negative, NP = not performed; NC = not conclusive).

ID Sample	Site	Date of Collection	PCR ^1^	PCR-RFLP *gyrB* Assay	Spoligotyping
1	Bertù	25 July 2017	N	NP	NP
2	Bertù	20 June 2017	N	NP	NP
3	Bocca di Lovere	25 July 2017	N	N	NP
4	Bocca di Lovere	20 September 2017	P	N	NC
5	Fa	25 July 2017	N	NP	NP
6	Fa	20 September 2017	N	NP	NP
7	Lama	26 July 2017	N	NP	NP
8	Lama	21 September 2017	N	NP	NP
9	Navone	25 July 2017	N	NP	NP
10	Navone	20 September 2017	N	NP	NP
11	Praa 1	25 July 2017	N	NP	NP
12	Praa 1 BIS	26 July 2017	N	NP	NP
13	Praa 2	25 July 2017	N	NP	NP
14	Praa 2	21 September 2017	N	NP	NP
15	Sembrune	26 July 2017	N	NP	NP
16	Sembrune	21 September 2017	N	NP	NP
17	Praa 3	25 July 2017	N	NP	NP
18	Praa 3	20 July 2017	N	NP	NP
19	Praa 3	20 September 2017	N	NP	NP

^1^ = IS6110 RT-PCR.

## Data Availability

The data presented in this study are available in the Appendix A section.

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
