# Peer review of "Mycobacterium microti at the Environment and Wildlife Interface"

_microorganisms, 2021, doi:10.3390/microorganisms9102084_

Round 1

Reviewer 1 Report

A well written paper describing a survey of M. microti in rodent tissue and environmental samples that were collected in 2016 and 2017 from an area of the central Italian Alps known to be affected by endemic M. microti infection in wild boar. The methodology is sound and described in detail in the paper and the conclusions are supported by the findings.  In my opinion, this work makes a significant contribution to our understanding of the epidemiology of this elusive organism of the MTB complex and it is worthy of publication in this journal.

I only have three minor suggestions for improving the style/terminology used in certain sections of the manuscript:

  1. In the Abstract (and elsewhere in the paper), please replace the ambiguous term "diffusion" with 'hotspot', 'clustering', 'high prevalence' or 'transmission', depending on the context of the sentence.
  2. Line 105 in Section 2.1 (Study Area) - I think that there is a typo here. Surely Lake Garda must be more than 65 metres above sea level. Should this not read "650 metres a.s.l."?
  3. In sections 2.4 and 2.5, please use superscripts to correctly describe the serial dilutions (concentrations) of M. bovis BCG used to spike the uncontaminated water and mud samples, i.e. 106, 105 and 101 CFUs etc. instead of 106, 105, 101.  Ditto for section 3.2 in the Results section. This may have arisen from a transcription error when preparing the galley proofs from the original manuscript.

Author Response

A well written paper describing a survey of M. microti in rodent tissue and environmental samples that were collected in 2016 and 2017 from an area of the central Italian Alps known to be affected by endemic M. microti infection in wild boar. The methodology is sound and described in detail in the paper and the conclusions are supported by the findings.  In my opinion, this work makes a significant contribution to our understanding of the epidemiology of this elusive organism of the MTB complex and it is worthy of publication in this journal.

We thank Reviewer 1 for the appreciation of our work.

I only have three minor suggestions for improving the style/terminology used in certain sections of the manuscript:

  1. In the Abstract (and elsewhere in the paper), please replace the ambiguous term "diffusion" with 'hotspot', 'clustering', 'high prevalence' or 'transmission', depending on the context of the sentence.

We thank Reviewer 1 for the suggestion and we replaced the term diffusion accordingly with more specific terms. In particular:

Line 16: high presence

Line 62: high presence

Line 70: transmission

Line 327: high presence

Line 330: transmission

Line 381: spread

Line 422: transmission

Line 458: transmission

  1. Line 105 in Section 2.1 (Study Area) - I think that there is a typo here. Surely Lake Garda must be more than 65 metres above sea level. Should this not read "650 metres a.s.l."?

We thank Reviewer 1 for this comment, but actually Lake Garda altitude is 65 m a.s.l. Lake Garda has been mentioned in the manuscript, together with Mount Caplone as the two altitudinal extremes of the study area which is located in the ‘Parco dell’Alto Garda Bresciano’.

  1. In sections 2.4 and 2.5, please use superscripts to correctly describe the serial dilutions (concentrations) of M. bovis BCG used to spike the uncontaminated water and mud samples, i.e. 106, 105 and 101 CFUs etc. instead of 106, 105, 101.  Ditto for section 3.2 in the Results section. This may have arisen from a transcription error when preparing the galley proofs from the original manuscript.

We thank Reviewer 1 for pointing that out and we apologize for that. We think the editing procedures may have caused this error. All the superscripts have been checked and corrected in sections 2.4, 2.5 and 3.2; moreover a general check has been performed throughout the whole manuscript.

Reviewer 2 Report

This manuscript is important for all of us living on the earth -- regarding the survival of the environment, wildlife, animals and humans. 

Minor comments:  Line 120. What does "accurate necropsies and dissection" mean here ? Do you mean "thorough, or complete, or recorded ?

Line 340. "microorganism." Lines 460-461.   Awkward wording for the second half of the sentence. Do you mean "--- and it identified a new species ( yellow-necked mouse) susceptible to this pathogen" .

Interestingly, a new relevant paper on the zoonotic bacterial and parasite presence in Central Italy was just published.  Please refer to it too in your paper.

Ebani, V.V.; Guardone, L.; Bertelloni, F.; Perrucci, S.; Poli, A.; Mancianti, F. Survey on the Presence of Bacterial and Parasitic Zoonotic Agents in the Feces of Wild Birds. Vet. Sci. 2021, 8, 171.  https://doi.org/ 10.3390/vetsci8090171

Author Response

This manuscript is important for all of us living on the earth -- regarding the survival of the environment, wildlife, animals and humans.

We thank Reviewer 2 for the very kind words of appreciation of our work.

Minor comments: 

Line 120. What does "accurate necropsies and dissection" mean here? Do you mean "thorough, or complete, or recorded ?

We thank Reviewer 2 for the comment. We removed the term ‘accurate’ which is not appropriate.

Line 340. "microorganism."

We thank Reviewer 2 for pointing that out, we corrected the mistake.

Lines 460-461.   Awkward wording for the second half of the sentence. Do you mean "--- and it identified a new species (yellow-necked mouse) susceptible to this pathogen".

We thank Reviewer 2 for the comment. We wanted to point out that in our study, a new rodent species has been found positive to this pathogen. We changed the sentence and now you can read: In this study, the presence of genotype SB2277 in a M. microti strain isolated from Mi-crotus sp. was demonstrated, moreover A. flavicollis was identified as a new susceptible species to this pathogen.

Interestingly, a new relevant paper on the zoonotic bacterial and parasite presence in Central Italy was just published.  Please refer to it too in your paper.

Ebani, V.V.; Guardone, L.; Bertelloni, F.; Perrucci, S.; Poli, A.; Mancianti, F. Survey on the Presence of Bacterial and Parasitic Zoonotic Agents in the Feces of Wild Birds. Vet. Sci. 20218, 171.  https://doi.org/ 10.3390/vetsci8090171

We thank Reviewer 2 for the comment. We read the suggested paper and found it very interesting. The role of birds as vectors of pathogens transmission through environmental contamination is definitively an under reported field of study. Ebani et al. investigated some wild avian species and no representative of the Mycobacterium genus was found. For this reason we think that at this stage citing Ebani et al will not give additional information to our manuscript.